# Analysis of Genes Associated with Feeding Preference and Detoxification in Various Developmental Stages of *Aglais urticae*

**DOI:** 10.3390/insects15010030

**Published:** 2024-01-03

**Authors:** Ouyan Xi, Wentao Guo, Hongying Hu

**Affiliations:** 1College of Life Science and Technology, Xinjiang University, Urumqi 830046, China; xoy@stu.xju.edu.cn (O.X.); 17699103800@163.com (W.G.); 2Xinjiang Key Laboratory of Biological Resources and Genetic Engineering, Urumqi 830046, China

**Keywords:** *Aglais urticae*, feeding preference, detoxifying enzymes, RNA-seq

## Abstract

**Simple Summary:**

*Aglais urticae* (Lepidoptera, Nymphalid) is a common butterfly distributed in temperate regions worldwide. The nettle *Urtica cannabina* (Urticaceae) is key to the survival of the small tortoiseshell, which provides its primary food source. This species is also a toxic plant listed in the Chinese Flora Atlas database, as it has poisonous roots and leaves. However, the phytophagous insect has evolved different adaptations against the plant′s defense system. This study aims to identify the genes linked to feeding strategies and adaptations of *Aglais urticae*’s larval feeding preference and detoxifying mechanism, and it may serve as an approach to help us understand the molecular process of its in vivo detoxification and find its related genes. Overall, these results provide the basis for future work on the detoxification and development of phytophagous insects by identifying the detoxification mechanism of *A. urticae* and their development-related genes in holometabolous insects.

**Abstract:**

Herbivorous insects and host plants have developed a close and complex relationship over a long period of co-evolution. Some plants provide nutrients for insects, but plants’ secondary metabolites also influence their growth and development. *Urtica cannabina* roots and leaves are poisonous, yet *Aglais urticae* larvae feed on them, so we aimed to clarify the mechanism enabling this interaction. At present, studies on the detoxification mechanism of the *A. urticae* are rare. In our study, first, we used the *A. urticae* larval odor selection behavior bioassay and choice feeding preference assay to analyze the feeding preferences of *A. urticae* on its host plant, *U. cannabina*. Next, we used transcriptome sequencing to obtain the unigenes annotated and classified by various databases, such as KEGG and GO. In this study, we found that *U. cannabina* could attract *A. urticae* larvae to feed via scent, and the feeding preference assay confirmed that larvae preferred *U. cannabina* leaves over three other plants: *Cirsium japonicum*, *Cannabis sativa*, and *Arctium lappa*. The activity of detoxifying enzymes GST and CarE changed in larvae that had consumed *U. cannabina*. Furthermore, through transcriptomic sequencing analysis, 77,624 unigenes were assembled from raw reads. The numbers of differentially expressed genes were calculated using pairwise comparisons of all life stages; the expression of detoxification enzyme genes was substantially higher in larvae than in the pupal and adult stages. Finally, we identified and summarized 34 genes associated with detoxification enzymes, such as UDP-glucose 4-epimerase gene, 5 Glutathione S-transferase genes, 4 Carboxylesterase genes, 4 Cytochrome P450 genes, 10 ATP-binding cassette genes, 4 Superoxide dismutase, and Peroxidase. Moreover, we identified 28 genes associated with the development of *A. urticae*. The qRT-PCR results were nearly consistent with the transcriptomic data, showing an increased expression level of four genes in larvae. Taken together, this study examines the correlation between *A. urticae* and host plants *U. cannabina*, uncovering a pronounced preference for *A. urticae* larvae toward host plants. Consistent with RNA-seq, we investigated the mechanism of *A. urticae*’s interaction with host plants and identified detoxification-related genes. The present study provides theoretical support for studying insect adaptation mechanisms and biological control.

## 1. Introduction

Herbivorous insects must overcome many obstacles when feeding on plants, such as the morphology and structure of plant leaves. However, different plant volatiles affect host plant selection in many species of lepidoptera, so insects and plants form a symbiotic relationship [1,2]. Generalists and specialists among herbivores can be distinguished based on their preferences for different plant types (including oligophagous and monophagous) [3]. The various responses to the secondary metabolites present in the plants mainly cause this variation. When an insect adapts to the secondary metabolites produced by a host plant, it may develop a parasitic or feeding relationship and process the secondary metabolites of host plants via detoxification and digestion, as well as sensory responses like taste and smell [4]. However, it is unclear whether the genes in an insect change when its behavior changes, so it is essential for us to explore the molecular mechanisms of this facet in more depth.

*Aglais urticae* L. (Lepidoptera and Nymphalidae), also known as small tortoiseshell, is a common butterfly worldwide. The main distribution areas include Scandinavia, Spain, southern Ireland, and Japan. Moreover, it exists everywhere except in northern Russia and India [5]. *A. urticae* is the specialist that feeds and oviposit on the host plant *Urtica* [6,7]. *Urtica* (Urticaceae) includes about 50 taxa and occurs primarily in temperate to subtropical regions such as Europe, North Africa, northwestern China, and Japan [8,9]. *Urtica cannabina* L. is a ubiquitous herbaceous perennial plant in North China. In Xinjiang (Northwestern China), the *U. cannabina* distribution zone is often formed between the lower mountain areas in the north and south of the Tianshan Mountains and the edge of Gobi. *U. cannabina* is a toxic plant. It contains phenols, lignans, and other nutrients, yet the stinging hairs on the leaves contain histamine, formic acid, and acetic and butyric acids. Contact with the stinging hairs causes a mildly painful sting, redness, and itching for at least a few minutes or days in humans [10,11,12]. However, *A. urticae* larvae frequently feed on *U. cannabina* (Figure 1A,B) [13,14]. As a specialist herbivore, *A. urticae* has been examined in conjunction with *Urtica dioica* to determine if it can overcome the nutritional restrictions imposed by its host plant. Additional studies have shown that larvae feeding on regrowth plants (the plants that had been cut after flowering and regrow) had better growth rates than mature plants, and pupal weights were higher. Much research has shown that *A. urticae* prefers *Urtica* plants. However, studies on its detoxification mechanisms are rarely reported [2,15,16].

A complex relationship exists between herbivorous insects and host plants, as plants resist insect feeding via various behaviors [17]. In addition to morphological barriers against herbivory, plants have developed specific chemical defenses against insects [18]. To better cope with the many toxic substances contained in the host leaves during feeding, phytophagous insects need to regulate the activities of a series of enzymes in their bodies to overcome the host’s defense mechanism to digest food better and ensure their normal development. For example, lepidopteran oligophagous insect larvae feed in large quantities on cotton leaves, and substances such as phenols and tannins contained in the leaves are toxic to insect larvae [4]. To resist the chemical defenses of plants, herbivorous insects have evolved several adaptation mechanisms of detoxification and metabolism [19,20]. Sumitha Nallu et al. found that by utilizing GWAS and RNA-seq in *Pieris rapae*, they identified several genes related to detoxification enzymes [21]. Carboxylesterase (CarE), Glutathione-S-transferase (GST), and Acetylcholinesterase (AChE) are the main detoxification enzymes in insects. In the genome assembly and developmental transcriptome of *Spodoptera exigua*, several genes are involved in digestion and detoxification, and the cytochrome P450s and UGTs are expressed explicitly in the larval stage [22,23]. Studies on the common cutworm *Spodoptera litura* found detoxification enzyme genes and insecticide-targeted receptor genes associated with detoxification and development [24]. When insects feed on host plants, secondary metabolic substances in plants activate or inhibit relevant detoxification enzyme activities, thus helping insect detoxification.

RNA-seq has been utilized extensively in many insect and plant interactions to study essential molecular mechanisms [25,26]. The behavioral mechanisms by which insects select their host plants have been studied to explain the association patterns observed in nature [23]. Researchers employed transcriptome assembly and gene expression analysis to control *Ostrinia furnacalis*. Transcriptomic analysis could help us study developmental pathways, wing formation, and olfactory recognition [27]. Extensive transcriptome data allowed researchers to uncover members of significant gene families linked with detoxification in *Grapholita molesta* [28]. RNA-seq is an important tool to help us understand the molecular mechanisms of insects that have evolved different adaptations to overcome plant defense systems. However, plant defenses and insect detoxification mechanisms remain unclear. Here, we hope to explore the defense mechanism of *A. urticae* larvae against *U. cannabina*.

Our research aimed to investigate the larval feeding performance of *A. urticae* on *Urtica cannabina* L. (host plants). We constructed and sequenced *A. urticae* transcriptome libraries, including three developmental stages (fourth-instar larvae, pupae, and adults) (Figure 1C), using a paired-end Illumina Novaseq6000 (Illumine, San Diego, CA, USA) sequencing platform. The differentially expressed genes (DEGs) between detoxification enzymes and developmental processes were identified. Furthermore, the annotation of gene function was based on seven databases. Taken together, these results may help elucidate the underlying detoxification mechanisms in different developmental stages of *A. urticae* and establish a foundation for further research on plant and insect interactions.

## 2. Materials and Methods

### 2.1. Study Sites and Insect Collections

The larval specimens were collected in Nanshan Mountain (Part of the eastern Tianshan Mountains in China) in Urumqi, Xinjiang, China (N: 43°24′55″, E: 87°18′46″, H: 1821.14 m), and the host plant was *U. cannabina*. Then, field-collected *A. urticae* larvae were individually placed in glass vials under a light incubator with temperatures of 26 ± 2 °C, 60 ± 5% RH, and a 14:10 h (L:D) cycle. *A. urticae* larval morphology is described as follows: the head is black, covered with long yellow and white hairs and small white spots. The body is covered with black hairs, the dorsal midline is fine and black, with yellow longitudinal stripes on both sides, and the ventral surface is yellow-green. The thoracic peduncle is black, and the ventral peduncle is yellow-green. Morphology does not differ significantly between ages except for body length. In our study, molting was observed from the first instar larvae and was used as a basis for the division of the instars. Next, one part of the larvae from each instar obtained from rearing was used for biological experiments, and the other part was kept in liquid nitrogen for subsequent experiments. RNA-seq samples were all obtained from *U. cannabina* and then reared in the lab.

### 2.2. A. urticae Larval Odor Selection Behavior Bioassay

The four-arm olfactometer was used to investigate larval feeding behavior. The angle between the four arms is 90°. The activity chamber, with a 30 cm diameter, 2 cm height, and four material bottles, was connected at the base of the four arms as the sample chamber. Fourth-instar larvae were randomly selected after 24 h of starvation (60 larvae per treatment), and the airflow was controlled at 150 mL/s. Four plants were selected from Nanshan Mountain: *U. cannabina*, *Cirsium japonicum*, *Cannabis sativa*, and *Arctium lappa* leaves (equal area size). They were placed in the material bottle and then aerated for 10 min to fill the pipe with odor and ensure accurate test results. The observation time for each larva was set at 8 min. If the larva crawled through 1/3 of the side wall and stayed for more than 1 min, it was marked as reacting to the volatile odor of the plant in the material bottle connected to the channel. Otherwise, it was marked as not reacting. The attraction rate of plant volatiles to larvae was calculated as follows:Attraction = Reaction larval number/The total number of larvae × 100%.

### 2.3. Larval Feeding Choices on Four Plants and Feeding Rates on U. cannabina Behavior Bioassay 

Determination of the feeding behavior of larvae on four plants (*U. cannabina*, *C. japonicum*, *C. sativa*, and *A. lappa*). (1) The fourth instar larvae were randomly selected and starved for 24 h. Leaves of four plants were selected from normally growing specimens and cut into five pieces of equal size (summing up to 1.413 cm^2^) in a petri dish, then diagonally placed; (2) The fourth instar larvae were placed in the middle of the petri dish and waited for the larvae to make a selection of the four plants to start feeding (first feeding). When this larva picked another plant, it was recorded as the second feeding. The time of each feeding was recorded. Larvae were equally divided into 3 groups (n = 10).

Studies on the rate of feeding on *U. cannabina*. We randomly selected second, third, and fourth instar larvae that were growing normally after 24 h of starvation treatment. Then, we used leaves from normally growing *U. cannabina* and leaves of the same growth stage to feed the larvae and observed different instar larvae feeding on leaves (equal size) to calculate the feeding area. The second, third, and fourth instar larvae were divided into 4 groups (5, 10, 15, and 20). The feeding time (with the end of the last larvae’s feeding being the final criterion) and feeding rate of each group were calculated. The process was repeated 3 times for each group. Finally, the feeding area, feeding time, and feeding rate of different larvae were analyzed.

### 2.4. Enzyme Activity Assays of Feeding on Different Plants

The carboxylipase and glutathione S-transferase activities were measured as follows. First, the fourth instar larvae were randomly selected and starved for 24 h. The larvae were allowed to feed on four plant leaves for 1 h. Post-feeding at 2 h, 4 h, and 6 h, the larvae were snap-frozen in liquid nitrogen. Next, the extraction of crude enzyme liquid was performed. The larvae, which fed on *U. cannabina*, *C. japonicum*, *C. sativa*, and *A. lappa* leaves at different times (2 h, 4 h, and 6 h), were processed under the condition of an ice bath. Liquid nitrogen was continuously added to grind the larvae to a powder, and the processed sample was transferred into a 1.5 mL tube. Crude enzyme extraction liquid was added, and following high-speed centrifugation, the supernatant was collected as the sample to be tested. Different centrifugation conditions were chosen according to the types of enzymes tested: carboxylipase at 15,000 rpm, 4 °C, and centrifugation for 10 min; Glutathione-S-transferase at 8000 rpm, 4 °C, and centrifugation for 10 min. Subsequently, the instructions for the activity assay kit (GST and CarE Activity Assay Kit, Solarbio, Beijing, China, BC0350 and BC0840) were followed, and the activity of both enzymes was calculated. Each group included three biological replicates.

### 2.5. Total RNA Extraction, De Novo Assembly, and Functional Annotation of Transcriptomic Data

The total RNA was extracted from the frozen *A. urticae* larvae (n = 3), pupae (n = 4), and adults (n = 4) in each group using the TRIzol reagent. According to the manufacturer’s instructions (Invitrogen, Carlsbad, CA, USA), the reagent and genomic DNA contamination were removed through DNase I (TaKaRa, Dalian, China) treatment to obtain highly purified total RNA. The quantity and integrity of RNA were assessed using the RNA Nano 6000 Assay Kit on the Bioanalyzer 2100 system (Agilent Technologies, Santa Clara, CA, USA). The library had to be tested to ensure its quality. After construction, the library was quantified using the Qubit2.0 Fluorometer (Thermo Fisher Scientific, Foster, CA, USA), then diluted to 1.5 ng/μL, and the insert size of the library was detected through the Agilent 2100 bioanalyzer (Agilent Technologies, Palo Alto, CA, USA). The effective concentration of the libraries was exactly determined using qRT-PCR to check the quality. 

Unigenes were annotated based on seven Databases (Nr, Nt, Pfam, KOG, Swiss-prot, KO, and GO). A model based on the negative binomial distribution was used in the Differential Expression Analysis of DESeq2 to provide statistical approaches for identifying differential expression in digital gene expression data [29]. The *p*-values were scaled for control by applying the method of Benjamini and Hochberg. Next, the false finding rate padj < 0.01 and log_2_(foldchange) > 1 was set as the criteria for significantly different expressions [30].

### 2.6. Identification and Analysis of Differentially Expressed Genes

GO seq (1.10.0) [31] and KOBAS (v2.0.12) [32] software were used for GO and KEGG pathway enrichment analysis of differential genes. It was based on the hypergeometric distribution principle. Differential genes were obtained through significant difference analysis and annotated to the GO or KEGG database. *A. urticae* larvae feed on leaves, while the pupal stage stops feeding, and the adults feed on the nectar of flowers. Therefore, we divided them into 3 groups for feeding: larvae were fed on *U. cannabina* leaves, adult females were fed on honey water (10%), and the pupal stage was not fed. In this study, we analyzed two groups to illustrate the detoxification enzyme genes: larvae versus pupae (L vs. P) and larvae versus adults (L vs. A).

Then, all stages, L vs. P, and P vs. A were compared to investigate the developmental genes in the two life stages. Reference sequences were provided for transcriptomes acquired using the Trinity platform, and the FPKM was applied to identify the number and expression level of each unigene. In the second instance, GO and KEGG pathway enrichment were used to analyze the DEGs. In this way, we obtained differentially expressed genes at different developmental stages. There has been no prior investigation of the transcriptome of *A. urticae*. This study is a preliminary analysis of DEGs during their developmental period.

### 2.7. qRT-PCR Analysis

To verify the results of the transcriptome analysis, we selected four genes related to detoxification enzymes for analysis: Cluster-4416.28326, Cluster-4416.47803, Cluster-4416.31951, and Cluster-4416.42797. β-actin served as the reference gene for qRT-PCR normalization. Primer sequences are listed in Appendix A. Reverse transcription of 1.5 μg of total RNA samples to cDNA was performed, and a 10-fold dilution of the cDNA sample was used as a template for the online assay. The total system volume was 20 μL (10 μL of SybrGreen qPCR master mix, 0.4 μL each of primers F and R, 7.2 μL of deionized water, and 2 μL of template cDNA). The temperature and time of the PCR cycle were as follows: initial steps at 95 °C for 3 min, followed by 45 cycles of 15 s at 95 °C and 30 s at 60 °C. The sample-spiked 96-well plate was placed in an ABI 7500 fluorescence quantitative PCR instrument for the reaction. Relative transcription levels were calculated using the double delta CT(2^−ΔΔCt^) method.

### 2.8. Statistical Analysis

The statistical analyses in this study were mainly based on SPSS 26 and GraphPad Prism8 software. One-way analysis of variance (one-way ANOVA) and Tukey’s multiple comparisons test at *p* < 0.05 were used to assess significant differences. RNA-seq analysis was applied to utilize the DESeq2 R package (1.20.0) and edge R package (3.22.5) [33].

## 3. Results

### 3.1. Analysis of the Host Plant Selection Behavior of A. urticae Larvae

The four-arm olfactometer was used to determine the odor selection of *A. urticae* larvae among four different plants. The time taken to select leaves from the four plants varied significantly (*p* < 0.001), with *U. cannabina* requiring less time compared to the other plants. The attraction rate of *U. cannabina* to larvae reached 78.3%. These results indicated that the leaf volatiles of *Urtica* were important in the host selection and location of *A. urticae* larvae (Figure 2A). The number of *A. urticae* larvae selected for *U. cannabina* leaves was significantly higher than the other three species (all *p* < 0.001). The lowest number of larvae chose *A. lappa* (Figure 2B). These results showed that *A. urticae* larvae preferred the volatile odor of *U. cannabina*.

### 3.2. Analysis of Feeding Preference of Larvae A. urticae among Four Plants

According to the choice feeding preference assay, *A. urticae* larvae preferred *U. cannabina* leaves. Many larvae chose *U. cannabina* when they first fed on plant leaves. However, some larvae were initially attracted to the other three plants, including *C. japonicum*, *C. sativa*, and *A. lappa*, for some time and then reselected *U. cannabina*. Finally, all test larvae chose to feed on *U. cannabina*, with a feeding rate of 100%. Under the controlled feeding area, the average feeding time for larvae was 1068.37 ± 230.55 s. In contrast, the time from the start to the first feeding of larvae on *U. cannabina* was significantly longer than in larvae feeding on control plants. The time when the larvae were first attracted to the plant leaves for feeding was designated as the first feeding behavior. The second feeding behavior refers to the feeding behavior of larvae that go on to select other plant leaves after the first feeding (Figure 2C). The results indicated that *A. urticae* larvae showed a clear feeding preference for *U. cannabina* and reflected the singularity of their feeding.

### 3.3. Analysis of the Feeding Rate at Different Instars with Different Densities of A. urticae Larvae

Our study compared the ability of *A. urticae* at different instars to feed on fresh *U. cannabina* leaves. The results showed that the larvae’s feeding rate and feeding area increased with the number of larvae in the same feeding area. The feeding rate and area of the larvae increased with the increasing number of larvae in the different instars. The fourth-instar larvae’s feeding time, area, and rate were significantly higher than those of second-instar larvae. Larvae feeding on *U. cannabina* were highly significant during the fourth instar (Appendix A). Under the four groups (n = 5, 10, 15, and 20) with different instars larvae (second, third, and fourth instar), the area and rate of feeding on *U. cannabina* increased with the increasing number of larvae. The feeding area and rate of larvae (n = 20) reached 16.801 ± 0.233 cm^2^ and 1.166 ± 0.051 mm^2^/s, respectively. The feeding rate of fourth-instar larvae did not significantly increase with 15 larvae, as also observed for 5 and 10 larvae (Figure 3).

### 3.4. Analysis of the Enzyme Activity Assay

In this study, compared to the other three plant species, *A. urticae* fourth-instar larvae feeding on *U. cannabina* showed the highest CarE enzyme activity after 2 h of starvation. After 4 h, CarE enzyme activity was higher in the larvae feeding on *U. cannabina* than in the other three plants. However, after 6 h, the enzyme activity in the larvae feeding on *U. cannabina* was higher than the other three plants. The results indicated that the activity of CarE was markedly higher in larvae feeding on *U. cannabina* than in the other three plants (*p* < 0.001). Furthermore, we investigated GST enzyme activity among all treatments. The analysis revealed significant differences in GST activity in the larvae feeding on *U. cannabina* compared to the other three host plants. It can be inferred that the secondary sexual substances released from *U. cannabina* could induce changes in GST activity in the larvae. However, GST enzyme activity increased in all larvae feeding on *U. cannabina*, *C. japonicum,* and *A. lappa*. In contrast, the lowest GST activity was observed in those feeding on *C. sativa*. The results showed that feeding on *U. cannabina* and *C. japonicum* was correlated with a difference in GST activity (*p* < 0.05). Additionally, feeding on *U. cannabina* and *C. sativa* was associated with a significant difference in GST activity (*p* < 0.001) (Figure 4).

### 3.5. Transcriptome Sequencing and De Novo Assembly

To identify genes that contributed to larvae detoxification and the development of the functions of *A. urticae* during metamorphosis, we extracted RNA from three developmental stages, including larva, pupa, and adult. Using transcriptomic sequencing, 28 million pieces of raw data were generated, and 27.2 million clean reads were obtained using the Illumina NovaSeq 6000 sequencing platform. The Q20, Q30, and GC content of the clean data were calculated. The means were 96.87%, 91.82%, and 43.14%; all samples had Q20 values exceeding 96%. In comparison, Q30 values exceeded 90%, confirming high sequencing accuracy. Finally, all downstream analyses were based on clean, high-quality data (Table 1).

Trinity software (v2.6.6) was used to assemble the clean reads for reference sequence analysis [34]. The unigene length ranged from 301 to 28,639 bp, with an average sequence length of 1208 bp. A transcript set of 77,624 unigene sequences with an N50 of 1999 bp was obtained (Table 2). Based on BUSCO software (BUSCO v5.4.7) analysis, 1079 complete single transcripts and 1837 multiple copies of transcripts were identified across all BUSCO groups [35].

### 3.6. Gene Functional Annotation

The sequences of unigene were annotated using BLASTX against KOG, protein family (PFAM), Swissprot, GO, KEGG, nucleotide sequences (NT), and non-redundant NCBI protein databases (NR). A total of 41,677 unigenes were annotated in more than one database. The NR database had the highest number of unigene annotations (30,550, 39.35%), and the least was KOG (11,543, 14.87%) (Table 3). When comparing *A. urticae* transcripts against the NR database, most annotated genes (14,050, 46.0%) were highly homologous to *Vanessa tameamea* (Lepidoptera: Nymphalidae) genes, followed by 8705 (28.5%) and 903 (3.0%) genes that were homologous to those of *Microplitis demolitor* and *Bicyclus anynana*, respectively. Another 24.60% of genes are homologous to other species. The results of the comparative analysis of homologous species are shown in Appendix A.

Gene annotation was conducted using three databases, including GO, KOG, and KEGG. After GO annotation, the successfully annotated genes were classified according to three major GO categories (BP, Biological process; CC, Cellular component; MF, Molecular function). The transcriptome had 46,933 (50.29%), 20,947 (22.45%), and 25,439 (27.26%) unigenes annotated through GO into BP, CC, and MF functional subclasses. Among the functional subclasses of biological process, the three most frequent processes were cellular process, metabolic process, and biological regulation; under cellular component were a cellular anatomical entity, intracellular, and protein-containing complex; and in molecular function was binding, catalytic activity, and transporter activity. The unigenes that KOG successfully annotated were classified based on the KOG database, which was divided into 26 groups. Of the unigenes that KOG successfully annotated, 1796 were only annotated for general function prediction (R), 1566 of which were annotated for signal transduction mechanisms (T), and 1167 were annotated for posttranslational modification, protein turnover, and chaperones (O). According to the KEGG database, 23,568 annotated unigenes were involved in 286 pathways, with signal transduction, transport, catabolism, and the endocrine system being the three most common pathways (Appendix A).

### 3.7. Analyses of Differentially Expressed Genes

We utilized DESeq2 software to perform pairwise comparisons of the three developmental stages. In the aggregate, 11,856 differential unigenes were found at the larval and pupal stages. Among them, 11,084 genes were upregulated, and 772 genes were downregulated in expression. Subsequently, there was a comparison of pupal to adult differential genes. Regarding 6223 differential genes, there were 1806 upregulated and 4417 downregulated genes. Finally, we also compared differential genes from larvae to adults and identified 13,127 genes, including 10,798 upregulated genes and 2334 downregulated genes. The analysis results were presented in volcano plots, with *p*(adj) < 0.01 and |log2(foldchange)| > 1 set as the threshold for significantly differential expression (Figure 5).

### 3.8. Screening for Detoxification Enzyme Genes

In the present study, we compared larvae versus pupae and larvae versus adults to identify differential genes associated with detoxification enzymes. The relevant signaling pathways were mainly obtained through GO and KEGG enrichment analysis (Figure 6). 

In the comparison of larvae versus pupae, we identified 7024 genes in 13 signaling pathways associated with detoxification enzyme genes, including the “protein binding”, “hydrolase activity”, “protein metabolic process”, “carbohydrate derivative binding”, “cellular amino acid”, “metabolic process”, “serine hydrolase activity”, “transcription cofactor activity”, “outer membrane”, “cellular homeostasis”, “histone acetyltransferase complex”, “proline metabolic process”, “reactive oxygen species metabolic process”, and “superoxide metabolic process”. Meanwhile, 29 genes associated with detoxification enzymes were also found, including UDP-glucose 4-epimerase gene, Glutathione S-transferase genes, Carboxylesterase (CarE) genes, Acetylcholinesterase (AchE) gene, Cytochrome P450 (P450) genes, and ATP-binding cassette (ABCs) genes. Secondly, two types of protective enzymes, including Superoxide dismutase (SOD), Peroxidase (POD), and Catalase (CAT) genes, were also identified. It was shown that all the related detoxification enzyme genes were upregulated in *A. urticae* larvae (Figure 7A). The larvae developed defense mechanisms against *U. cannabina* resistance in two ways. The first was through antioxidant enzymes, including SOD, POD, and CAT, that promote nutrient digestion and resistance to harmful substances. The second was through the digestive tract and fat body, which could secrete various detoxifying enzymes such as P450s, GSTs, AchEs, and CarEs to defend against plant secondary metabolites. To gain more insight into the potential functions of DEGs, enrichment analysis of KEGG pathways revealed pathways involved in detoxification and metabolism, such as “Glutathione metabolism”, “Amino sugar and nucleotide sugar metabolism”, “Metabolism of xenobiotics by cytochrome P450”, and “Drug metabolism cytochrome P450”. In addition to the genes analyzed through GO, Alcohol dehydrogenase 5 (ADH5), CYP3s, cathepsin B (cat-B), SUMF1, and Heat shock protein (HSP) genes were also found to be involved in the detoxification process in the signaling pathway of “Drug metabolism cytochrome P450” and “Insect hormone biosynthesis”. CYP3, HSP, cat-B, and ADH5 genes are essential in detoxifying exogenous toxic substances [36].

In the comparison of larvae versus adults, we identified 12 signaling pathways involving 9097 DEGs linked with detoxification and digestion. DEGs were most prevalent in “catalytic activity”, “single-organism localization”, and “intracellular organelle content”. However, in the comparison between larvae and pupae, the three signaling pathways with the highest enrichment of DEGs were the biological process “protein metabolic process” and molecular function “protein binding” and “hydrolase activity”. The major genes included UDP-glucose genes, P450 genes, GST genes, AchE genes, CarE genes, SOD genes, CAT genes, and ABC genes (Figure 7B). However, there was still a difference in the number of detoxification enzyme-related genes compared to the L versus P. For the KEGG analysis, there were 507 differentially expressed genes using KEGG that showed enrichment across 12 pathways. The signaling pathways with the most differential genes were “oxidative phosphorylation” and “lysosome”. The analysis was summarized, and the gene names and IDs of genes were analyzed using KEGG enrichment in this study (Appendix A).

### 3.9. Gene Expression Analyses across the Development Stages of A. urticae

To find differentially expressed genes (DEGs) during *A. urticae* development, we performed comparisons of three developmental stages. Pairwise comparisons were conducted among the larval to pupal period and pupal to adult developmental stages using the pathways, annotations, and study of genes that regulate insect growth and development. A total of 28 genes linked to insect development were identified.

During larval and pupal development, the top 10 signaling pathways with the most differential genes were selected for GO enrichment analysis. The GO term was significantly enriched for the metabolic process. The DEGs were highly associated with chitin binding in the molecular function (MF) group. Regarding cellular components, these genes were mainly enriched in DNA-directed RNA polymerase II and nuclear part-related pathways. Among them, eight GOs were enriched in CC, the highest number among these three groups (Table 4). The most upregulated genes were related to the nuclear part, including E3 ubiquitin-protein ligase CHIP, homeotic protein antennapedia, N-alpha-acetyltransferase, rho GTPase-activating protein 20-like, asteroid, glycine-rich cell wall structural protein 1.8-like, segmentation protein cap’n’collar-like, and apolipophorin III. The upregulation of chitinase 3 mainly characterized the chitin-binding pathway. The most downregulated genes included cuticle protein 19.8 isoform X1, cuticle protein 3, and calcium-binding protein P-like isoform X1, while the top 20 pathways were summarized using KEGG enrichment (Figure 8A). Through KEGG enrichment pathways, the basic upregulated pathways were “protein processing in endoplasmic reticulum” and “ubiquitin-mediated proteolysis”. The most downregulated pathways included “peroxisome” and “tight junction”; the significant DEGs was MYH6.

Comparing pupal and adult transcriptomes, the top 10 signaling pathways were mainly involved in serine-type activity, oxidation process, and chitin/iron ion/chitin binding. The GO term was significantly enriched for the oxidation–reduction process (Table 5). In the pupal stage, highly expressed genes were cuticle protein 8-like, pupal cuticle protein 36-like (PCP), serine proteinase stubble, and chitinase 2. In contrast, highly expressed genes in adults were endocuticle structural glycoprotein SgAbd-8-like, carboxypeptidase N subunit 2-like, fatty acyl-CoA reductase wat-like (FAR), chymotrypsin-C-like, trypsin-7-like, serine protease 7-like, and inositol oxygenase-like. Analysis of the KEGG enrichment pathway revealed the two major upregulated signaling pathways as the “Hippo signaling pathway” and “Curtin, suberin, and wax biosynthesis”.

In addition, the two main downregulated signaling pathways were “taste transduction” and “vitamin digestion and absorption”. The present study also identified important pathways related to female adult development of “ovarian steroidogenesis” and “olfactory transduction”. The major related genes included adenylate cyclase 3 (ADCY3), cyclic nucleotide-gated channel subunit alpha 3 (CNGA3), and CaM kinase II (CAMK2). The top 20 pathways were summarized using KEGG enrichment (Figure 8B).

### 3.10. Identification of the Differentially Expressed Genes (DEGs)

To validate the genes screened through RNA-seq, we chose four upregulated genes associated with detoxification enzymes and protecting enzymes for qRT-PCR analysis. The results showed that the differentially expressed genes screened were partially consistent with the RNA-seq data. The results in FPKM values and qRT-PCR indicated that four upregulated genes are expressed at higher levels in larvae than in pupae and adults. However, CarE expression was low relative to qRT-PCR, while GST expression was increased relative to qRT-PCR. SOD and CAT exhibited similar trends. The qRT-PCR results showed that four upregulated genes were highly expressed in larvae, indicating the reliability of the transcriptome data (Figure 9).

## 4. Discussion

In the present study, we found the feeding preferences by selecting the main plant taxa in the area where *A. urticae* larvae inhabit *U. cannabina*, *C. japonicum*, *C. sativa*, and *A. lappa*. In the field, it was found that *A. urticae* only laid their eggs on the abaxial surface of *U. cannabina* leaves, indicating that *A. urticae* have a clear preference for *U. cannabina*.

The attraction rate of *U. cannabina* to larvae in selecting different plant odor sources was 78.3%, while it reached 100% for feeding. The reason for this difference may be that during the process of feeding on plant leaves, the larvae’s smell and taste combined to determine their final selection. Only 78.3% of the test larvae responded to *U. cannabina* because they depended only on their sense of smell to choose the plant taste source throughout the selection phase of various volatile odors of plants. When the larvae used their receptors, *U. cannabina* was specifically attracted, which was chosen over other sources of plant volatiles. Therefore, the volatiles released from *U. cannabina* could be further collected and analyzed using gas chromatography–mass spectrometry (GC–MS) and electroactive potential (EAG) techniques to determine the types and contents of volatiles [37].

Based on measurements of feeding time, area, and rate of individual larvae at different instars, it was shown that, for *A. urticae*, the feeding area and rate of the fourth instar larvae were much higher than those of the second and third instar. The size of individual larvae significantly varied. The fourth instar larvae need much more food to produce sufficient energy. The feeding ability of the larvae increased with their growth. We found that the average feeding area of larvae in groups was lower than that in single experiments because the intra-species competition during the group feeding process resulted in the more vulnerable larvae feeding less [38]. Although we did not know how effective the control effect was, during the field survey, we discovered that *A. urticae* larvae had a robust controlling influence on the germination of *U. cannabina*. In the current study, we used different numbers and instars to evaluate the feeding rate of *U. cannabina* and determine its larval stage and population. 

Plants employ a range of secondary metabolites to protect themselves against phytophagous insect feeding, either preventing phytophagous insects from feeding or by inducing avoidance of feeding. On the other hand, insects have increased their adaptability to host plants over a long period of co-evolution by increasing their detoxification enzyme activity and upregulating the expression of detoxification enzyme genes to defend themselves against secondary compounds. The most important detoxification enzyme systems play a major role in managing plant secondary metabolites. Based on the findings of this investigation, it was postulated that CarE and GST could potentially serve as mechanisms for detoxifying secondary metabolizable substances in *U. cannabina* following consumption by *A. urticae* larvae. Examining detoxifying enzymes can offer new insights for the subsequent implementation of biological pest control measures and the investigation of insect resistance.

Secondary metabolites are usually classified according to their biosynthetic pathways. In general, three major molecular families include phenolics, terpenes, and steroids; alkaloids; and flavonoids [39]. Herbivores can adapt to these compounds through changes in sensory genes and detoxification or sequestration of toxic metabolites [40]. Among herbivores, secondary metabolites of plants lead to increased expression of detoxification enzymes in insects. P450s are members of an important superfamily of metabolic enzymes [24]. GSTs are multifunctional enzymes involved in responding to xenobiotics and oxidative stress. CarEs have a variety of physiological functions in insects, including dietary detoxification. CarEs play a crucial physiological role in detoxifying xenobiotics [41,42,43]. In addition to the four genes mentioned, UGTs and ABCs also have key roles in detoxifying secondary metabolites of plants. The ABC transporter superfamily, involved in xenobiotic transport and detoxification, plays a vital role in several insects [44], such as *Helicoverpa armigera* [45], *Diaphorina citri* [46], and *Bemisia tabaci* [47]. Analysis through RNA-seq revealed that in monarch larvae feeding on several types of milkweeds, a large number of DEGs are associated with detoxification in larvae, for instance, P450s, UGTs, ABCs, and GST [48,49]. Furthermore, the host plant’s secondary metabolites and defense enzymes cause a metabolic stress response in the insect body, elevating protection enzyme levels and speeding up the detoxifying metabolism of plant secondary metabolites to prevent or lessen harm.

The main protective enzymes include SOD, ROS, POD, and CAT [50,51,52]. Insect tolerance to plant secondary metabolites can be enhanced by detoxifying and using protective enzymes [53,54]. Several studies have reported that fresh *Urtica* leaves contain phenolic acid, tannins, and flavonoids. *Urtica* is an important medicinal plant, and its extracts can be used as a biological pest control agent due to the abundance of plant secondary metabolites [55,56,57]. The current findings show that the expression of detoxifying and protective enzymes in *A. urticae* larvae, after feeding on *U. cannabina*, is significantly higher than in pupal and adult instars. The larval stage genes are highly expressed because lepidopteran insects feed on host plants primarily as larvae. Since *A. urticae* are monophagous insects [58], the influence of secondary metabolites on plant growth and development is diminished by the increased production of detoxifying enzymes in their bodies. This can be recognized through transcriptome sequencing, focusing on the molecular mechanisms of feeding, digestion, and detoxification, and then identifying detoxification-related genes and protective enzyme genes in *A. urticae*. Insect adaptation and survival are dependent on feeding. In our study, we found a broad transcriptomic response involving digestion, immunity, and cuticular processes. From an applied viewpoint, identifying the mechanisms underlying adaptation to host plants in insects could yield new insights for developing sustainable pest control strategies.

*A. urticae* is an insect that undergoes metamorphosis. Differential genes associated with development were identified through transcriptome sequencing, comparing developmental stages. During the larvae’s growth and developmental stages, the most relevant signaling pathways include the nuclear part and chitin-binding pathway. E3 ubiquitin-protein ligase CHIP and rho GTPase-activating protein 20-like are crucial in cell cytoskeletal organization, growth, differentiation, neuronal development, and synaptic functions [59,60]. Homeotic protein antennapedia is expressed along the embryo’s anterior–posterior axis and participates in defining the segmented pattern of the embryo [61]. Asteroid gene function is mainly related to eye development and is essential in embryogenesis [62]. Segmentation protein cap’n’collar-like is highly expressed in larvae and is associated with the development of body segments in larvae. For example, the segment genes in *Drosophila* larvae are connected to establishing naked-cuticle and denticle belt areas [63], and chitin biosynthesis is necessary for insect molting and development. Meanwhile, chitinase 3 and N-alpha-acetyltransferase are associated with the development of chitin. The primary PCP expressions in synthesis in *A. urticae* pupae occur at pupation and are correlated with a change in the morphological characteristics of the cuticular lamellae [64,65,66]. During the adult stage, insects require large amounts of energy and have enhanced metabolic activity. The key genes involved include carboxypeptidase N subunit 2-like, fatty acyl-CoA reductase wat-like (FAR), chymotrypsin-C-like, trypsin-7-like, serine protease 7-like, and inositol oxygenase-like. The “ovarian steroidogenesis” and “olfactory transduction” pathways were also identified. The related genes include ADCY3, CAMK2, and CNGA3. Adenylate cyclase type 3 (ADCY3) increased cAMP levels during olfactory transduction and was associated with signal transduction [67]. CaM-kinase II(CAMK2) has high expression of CAMK2 in the glomeruli’s neuropil and all antennal lobe neurons of *Manduca sexta* [68,69,70]. CNG channels are composed of alpha and beta subunits. The CNG channel consists of α and β subunits and is associated with rat olfactory neurons [71]. Cyclic nucleotide-gated channel subunit alpha 3 (CNGA3) may have a role in the development of olfactory nerves [72,73]. Thus, these three genes are involved in developing *A. urticae*’s olfactory system. Lepidoptera (Nymphalid) are holometamorphic insects. Their ability to feed and move varies greatly at different stages of development. Using transcriptome sequencing to study the gene expression and regulatory mechanisms at different stages of their growth will be useful for understanding the changes and differences of genes throughout the development process and provide a molecular basis for the subsequent control of harmful insects and the protection of beneficial insects.

In this research, we used four dominant local plants to study the feeding preferences of larvae. We investigated the host adaptation of *A. urticae* to its host plant and analyzed larvae feeding on *U. cannabina*. The results provided basic data to understand the feeding preferences of phytophagous insects. Further, a comprehensive and high-quality transcriptome of *A. urticae* from the larva, pupa, and female adult was sequenced to provide a quality genetic database for this species. In this study, we identified several key genes associated with the underlying detoxification mechanisms and development of *A. urticae*. Hence, our results provide detailed information about the genetic data of *A. urticae*, which might be crucial for future studies. As it is a non-model insect, relatively few studies have been conducted on the molecular mechanisms of *A. urticae*; fortunately, our current findings provide a reference for subsequent investigations on Nymphalidae. Plants and phytophagous insects have complex chemical interactions, and studying the molecular mechanisms of their interactions can provide a preliminary understanding of the interactions at the molecular level, as well as an essential theoretical basis and data for developing new pest prevention and pest-control technologies. However, in our study, only *U. cannabina* was studied, and there are many species in the *Urtica*. In the future, our group will also explore other species of the genus *Urtica* in Xinjiang to determine the feeding of *A. urticae* larvae on other species. We considered expanding the field release of larvae to control *U. cannabina* because we found that *Urtica* thrives in the wild fruit forests of western Tianshan in Xinjiang, encroaching on the ecological niches of other vegetation and leading to a disruption of the ecological balance. We hope that in the future, we can apply detoxification enzyme genes to the biological control of pests, as well as use them in agricultural production.

## Figures and Tables

**Figure 1 insects-15-00030-f001:**
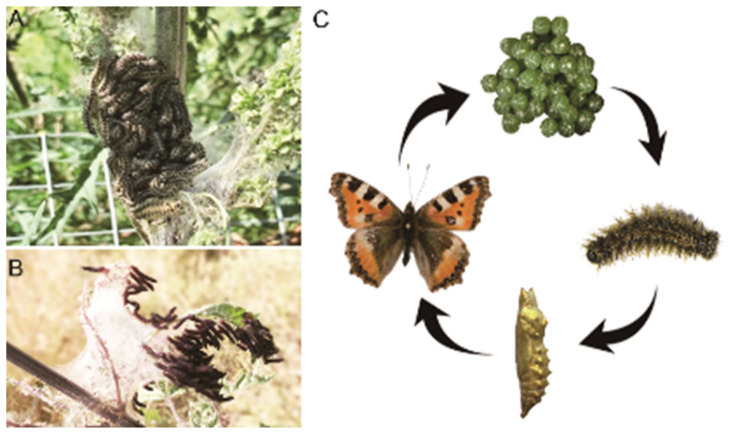
Developmental stages of *A. urticae*. (**A**,**B**) Larval foraging behavior; (**C**) Life cycle of *A. urticae*, including four developmental stages: egg, larva, pupa, and adult worm form.

**Figure 2 insects-15-00030-f002:**
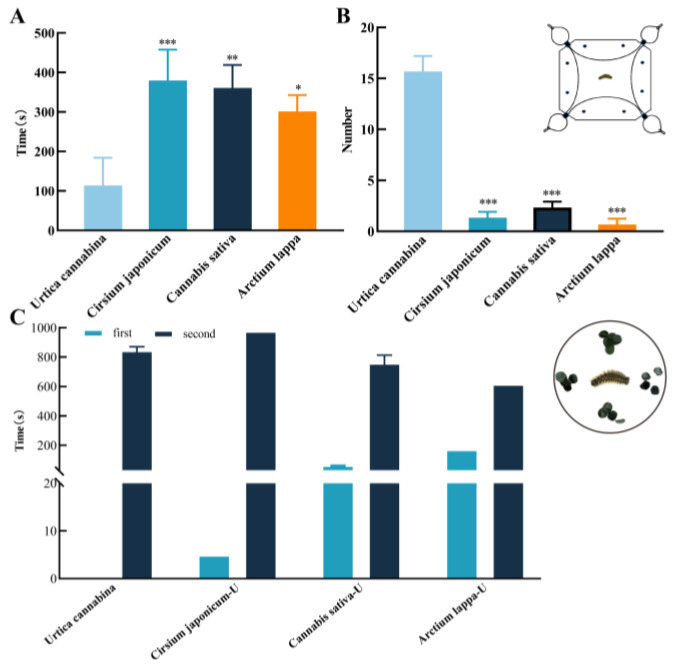
*A. urticae* larvae feeding preference. (**A**) Attraction of four plant odors to larvae, with the vertical axis representing the time of selection of the larvae for the odor source. (**B**) Number of larvae attracted to four plant odors, using a four-arm olfactometer, is shown on the vertical coordinate. (**C**) Larval feeding choices, where the first represents the time at which the larvae were attracted to feed at the beginning (blue columns); the second represents the time of feeding for a period and then re-selecting another plant for feeding (dark blue columns); the vertical axis represents feeding time. * indicates a significant difference (*p* < 0.05); ** indicates highly significant differences (*p* < 0.01); *** indicates highly significant differences (*p* < 0.001), respectively.

**Figure 3 insects-15-00030-f003:**
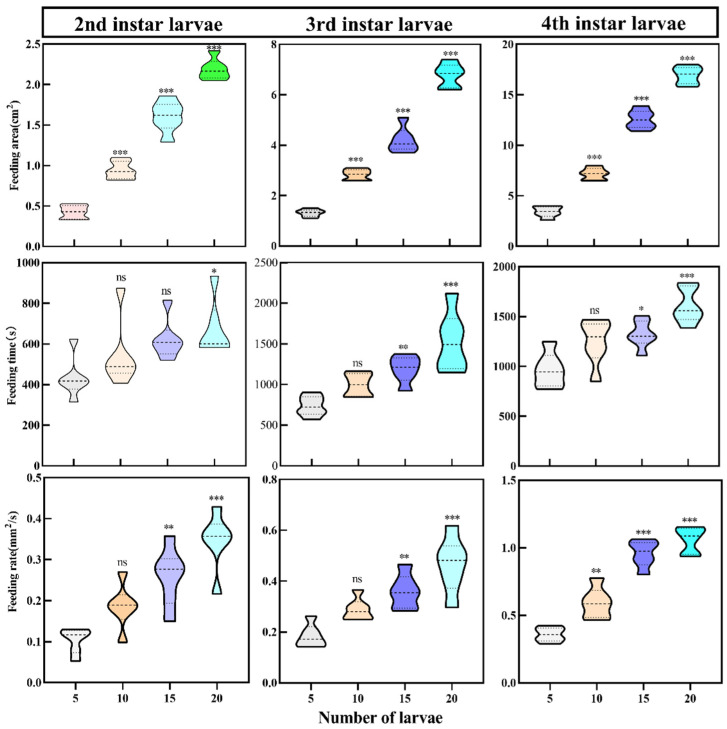
Leaf area, rate, and time of larval feeding at different *A. urticae* instars. Comparison of feeding area, feeding rate, and feeding time of larvae at different instars and numbers. The horizontal coordinate stands for the number of larvae. Comparison of *A. urticae* larvae between second instar larvae, third instar larvae, and fourth instar larvae and different numbers (5, 10, 15, and 20). The vertical coordinates indicate feeding area, feeding time, and feeding rate, respectively. “ns” indicates insignificant difference; * indicates a significant difference (*p* < 0.05); ** indicates highly significant differences (*p* < 0.01); *** indicates highly significant differences (*p* < 0.001), respectively.

**Figure 4 insects-15-00030-f004:**
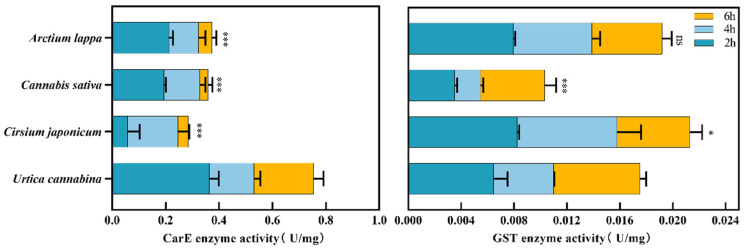
CarE and GST detoxifying enzymes of larval feeding in *A. urticae* larva. The vertical axis is for the four plants, and the horizontal axis indicates enzyme activity. Dark blue represents 2 h after fetching, light blue is 4 h, and orange represents 6 h of detoxifying enzyme activity. One-way ANOVA and Tukey’s multiple comparisons test at *p* < 0.05 were used to assess significant differences. “ns” indicates insignificant difference; * indicates a significant difference (*p* < 0.05); *** indicates highly significant differences (*p* < 0.001), respectively.

**Figure 5 insects-15-00030-f005:**
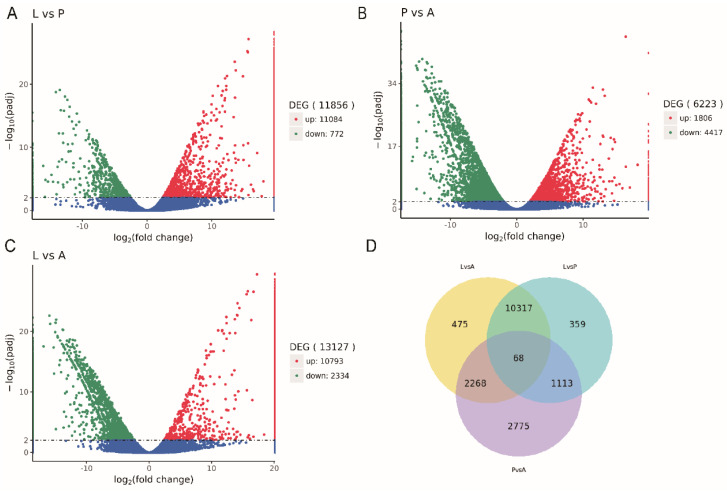
Volcano plots and Venn diagram of the differentially expressed genes identified in the larvae, pupae, and adults. (**A**) L vs. P; (**B**) P vs. A; (**C**) L vs. A; (**D**) Venn plots. Blue splashes indicate genes with no significant differential expression, red splashes indicate significantly upregulated genes and green splashes indicate significantly downregulated genes. Venn diagram of the number of differentially expressed genes in the L vs. P, P vs. A, and L vs. A. L: larvae; P: pupae; A: adults.

**Figure 6 insects-15-00030-f006:**
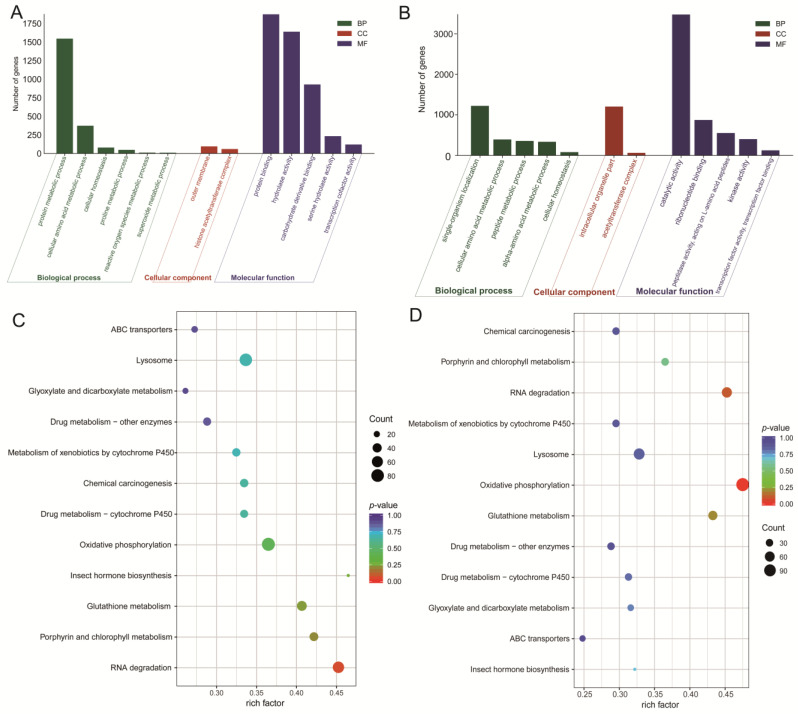
GO classification and KEGG enrichment pathway analysis of DEGs of detoxification enzyme genes at different developmental stages in *A. urticae*. (**A**) L vs. P; (**B**) L vs. A. GO pathways were summarized into three main categories: BP, CC, and MF. The vertical axis is the number of unigenes and GO terms identified in the larva, pupa, and adult. (**C**) L vs. P; (**D**) L vs. A. KEGG pathways with enriched annotations were shown in a scatterplot for DEGs. The rich factor indicates the ratio between the number of DEGs in a pathway and the total number of genes. The colored stripe indicates the range of *p*-value, and the size of the dots are gene numbers, respectively (*p* < 0.05).

**Figure 7 insects-15-00030-f007:**
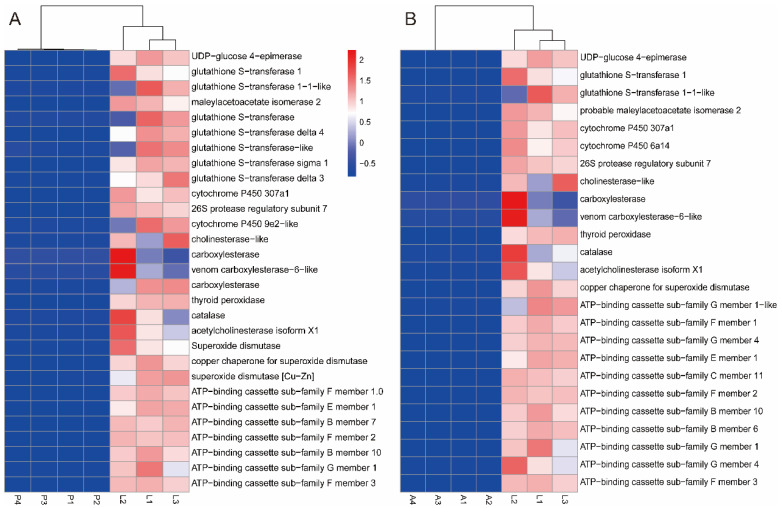
Expression heatmaps for differentially expressed detoxification enzyme genes of *A. urticae*. (**A**) L vs. P detoxification enzyme genes; (**B**) L vs. A detoxification enzyme genes. A hierarchical clustering was performed using Log 2 (fold change) scaled expression FPKM values to calculate the relative FPKM value for each gene.

**Figure 8 insects-15-00030-f008:**
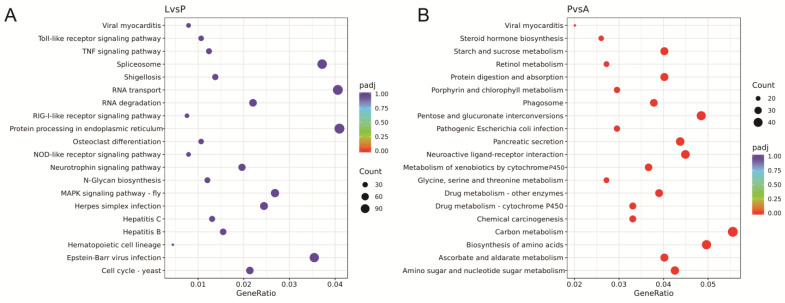
The top 20 significantly enriched KEGG pathways of the DEGs between different developmental stages of *A. urticae*. (**A**) L vs. P; (**B**) P vs. A. The x-axis label shows the rich factor. The gene ratio represents the number of DEGs/total number of genes in the KEGG pathway. The y-axis label shows the KEGG pathways. The color of the dots represents the *p*-value, and the size of the dot represents the number of DEGs enriched in the pathway. L: larvae; P: pupae; A: adults.

**Figure 9 insects-15-00030-f009:**
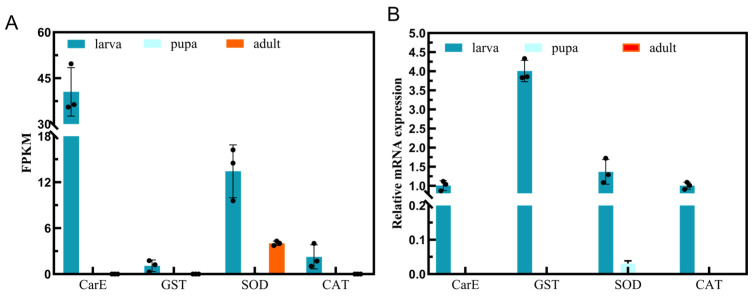
Validation of the differentially expressed genes using qRT-PCR. (**A**) RNA-seq value; (**B**) qRT-PCR. It was calculated using the 2^−ΔΔCt^ method. The data represent means ± standard deviation (SD) from three biological replicates. The RNA-seq value was based on the fold change of upregulation DEGs. CarE, GST, SOD, and CAT genes related to detoxification. Larva, pupa, and adult were different instars of *A. urticae*. The black dots represent data scatter.

**Table 1 insects-15-00030-t001:** Summary of *Aglais urticae* larva (L), pupa (P), and adult (A) sample sequencing data quality.

Sample	Raw Reads	Clean Reads	Error Rate (%)	Q20 (%)	Q30 (%)	GC (%)
L1	23,070,192	22,090,523	0.03	96.38	90.67	43.24
L2	23,274,479	22,478,655	0.03	96.37	90.98	41.92
L3	23,030,626	22,233,626	0.03	96.93	91.87	42.48
P1	22,264,683	21,031,052	0.03	97.29	92.5	43.52
P2	22,524,081	21,926,996	0.03	97.26	92.43	45.26
P3	22,754,615	22,329,792	0.03	97.15	92.35	44.34
P4	23,235,468	22,936,653	0.03	96.81	91.82	45.27
A1	23,843,502	23,226,075	0.03	96.2	90.66	42.19
A2	49,362,889	48,037,771	0.03	97.55	93.21	42.53
A3	23,102,754	22,541,196	0.03	96.9	91.92	41.39
A4	23,441,887	22,809,703	0.03	96.72	91.61	42.41

**Table 2 insects-15-00030-t002:** Assembly statistics of the *Aglais urticae* transcriptome.

Type	Min Length	Mean Length	Median Length	Max Length	N50	N90	Total Nucleotides
Transcripts	301	1376	828	28,639	2183	562	267,694,552
Unigenes	301	1208	666	28,639	1999	469	93,760,628

**Table 3 insects-15-00030-t003:** Statistics on the success rate of gene annotation in seven databases.

Database	Unigenes	Percentage (%)
Annotated in NR	30,550	39.35
Annotated in NT	32,045	41.28
Annotated in KO	14,361	18.5
Annotated in SwissProt	19,479	25.09
Annotated in PFAM	21,916	28.23
Annotated in GO	21,914	28.23
Annotated in KOG	11,543	14.87
Annotated in all Databases	6804	8.76
Annotated in at least one Database	41,677	53.69
Total Unigenes	77,624	100

**Table 4 insects-15-00030-t004:** Top 10 most specific GO terms enriched between larvae and pupa of genes related to the development of *A. urticae*.

GO ID	GO Name	GO Category	Up	Down	*p*-Value
GO:0005672	transcription factor TFIIA complex	CC	60	5	2.01 × 10^−11^
GO:0090575	RNA polymerase II transcription factor complex	CC	68	5	1.92 × 10^−10^
GO:0044798	nuclear transcription factor complex	CC	69	5	4.61 × 10^−10^
GO:0016591	DNA-directed RNA polymerase II	CC	76	5	9.06 × 10^−10^
GO:0019538	protein metabolic process	BP	1499	47	1.31 × 10^−9^
GO:0000428	DNA-directed RNA polymerase complex	CC	87	5	1.44 × 10^−9^
GO:0055029	nuclear DNA-directed RNA polymerase complex	CC	87	5	1.44 × 10^−9^
GO:0008061	chitin binding	MF	60	18	7.51 × 10^−9^
GO:0044428	nuclear part	CC	477	14	1.54 × 10^−8^
GO:0000124	SAGA complex	CC	41	0	2.94 × 10^−8^

**Table 5 insects-15-00030-t005:** Top 10 most specific GO terms enriched between pupa and adult of genes related to the development of *A. urticae*.

GO ID	GO Name	GO Category	Up	Down	*p*-Value
GO:0004252	serine-type endopeptidase activity	MF	95	22	5.11 × 10^−17^
GO:0008236	serine-type peptidase activity	MF	106	24	7.49 × 10^−13^
GO:0017171	serine hydrolase activity	MF	106	24	7.49 × 10^−13^
GO:0008061	chitin binding	MF	15	32	5.11 × 10^−12^
GO:0042302	structural constituent of cuticle	MF	7	44	5.23 × 10^−12^
GO:0016491	oxidoreductase activity	MF	208	70	1.58 × 10^−10^
GO:0055114	oxidation-reduction process	BP	209	69	3.19 × 10^−9^
GO:0005506	iron ion binding	MF	60	8	7.93 × 10^−9^
GO:0016705	oxidoreductase activity, acting on paired donors, with incorporation or reduction of molecular oxygen	MF	51	11	1.65 × 10^−7^
GO:0020037	heme binding	MF	50	9	5.36 × 10^−7^

## Data Availability

All other data presented in this study are available within this manuscript and its Appendix A. All raw RNA-seq reads data can be accessed at NCBI SRA. Accession: SAMN37395693, SAMN37395694, and SAMN37395695 (*A. urticae* larva); SAMN37395689, SAMN37395690, SAMN37395691, and SAMN37395692 (*A. urticae* pupa); SAMN37395685, SAMN37395686, SAMN37395687, and SAMN37395688 (*A. urticae* adult).

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
