# Peer review of "Analysis of Genes Associated with Feeding Preference and Detoxification in Various Developmental Stages of Aglais urticae"

_insects, 2024, doi:10.3390/insects15010030_

Round 1
Reviewer 1 Report
Comments and Suggestions for Authors
The authors wrote an interesting manuscript titled: Analysis of Genes Associated with Aglais urticae Feeding Preference and Detoxification Enzyme in Its Different Development Stages. They obtained RNA-seq data from different developmental stages of a phytophagous insect and describe the gene expression of genes associated to detoxification mechanisms.
The work is relevant, however there are some parts of the manuscripts that are not clear regarding the experimental design, please see below for each of the comments. Also some whole sections of the manuscript are not complete linked to the whole study, maybe the authors can include the rationale for including them in this work, for example, the section 2.3. Feeding Preference of A. urticae Larvae Towards Four Plants, the authors only did RNA work using one plant, right? Was this experiment done to select into which plant to focus the RNA work? Or are the authors planning to do RNA work using the other plants as hosts too? The purpose of this and relevance to the current work is not clear.
Simple summary: The authors wrote: “This study aims to identify the genes 13 linked to feeding strategies and adaptations of these nymphalid larvae to U. cannabina defensive mechanism to comprehend A. urticae larval feeding preference and the detoxifying mechanism at the molecular level.” Can the authors split the statement in two sentences and use common language to a broader audience?
Abstract: The first time that the authors mention a species name, please spell the whole species name and do not abbreviate it.
Line 21: the authors wrote: “three other plants”, which were the three other plants? Please list them in the abstract.
Line 22: “Next, RNA-seq has been widely used to investigate insect molecular mechanisms.” Could be deleted.
Line 23: “differential genes” do the authors mean: “differential gene expression”?
Line 26: “numerous” or put the number or delete
Line 27: N50 of 1,999 bp. Can the authors include an additional quality metric beside scaffold N50, for example BUSCO? See this article for further discussion: https://onlinelibrary.wiley.com/doi/full/10.1111/1755-0998.13364
Introduction
Lines 71-72: “Larvae feeding on regrowth plants showed better growth rates than mature plants, and pupal weights were also higher” can the authors explain the what regrowth plants mean for non-plant researchers?
Line 125: 2.1. Study Sites and Insect Collections, please add plant host from where the larvae were collected. Also add guidelines used to make the morphological determination of the larva development and to differentiate 4th larvae from early larvae stages.
Line 168-169: In the section, 2.5. Total RNA Extraction, De novo Assembly and Functional Annotation of Transcriptomic Data, please include number of individuals from each developmental stages that were used for the RNA extractions, and if they were collected in the field from the same pant host.
Line 194: why did the authors use honey water and not sugar water? Is there a reference to cite regarding this approach? Does honey have any type of enzyme or nutrients that may also affect detoxification?
Line 253: The material and methods for the section of results explained in “3.3. Analysis of the Feeding Rate at Different Instars with Different Densities of A. urticae Larvae” is missing, please add how the authors characterize each larvae stage and the methods for this result section. It is confusing because in methods it seems that the authors only used 4th larvae, pupae and adult, but here 2nd and 3rd larvae were also used. Also, what was the purpose of this part of the research? What was the question that the authors wanted to answer? Wouldn’t you expect 4th larvae to eat more than 2nd larvae? Or were some reasons to think otherwise?
Line 275: In the section: 3.4. Analysis Enzyme Activity Assay, please specify which larvae stage was used.
Line 471-473: The authors wrote: “The results showed that the differential genes screened were consistent with the results of RNA-seq, and that the four up-regulated detoxification enzyme and protecting enzymes genes were indeed highly expressed in A. urticae larvae (Fig. 9).” However, the results showed the same pattern, but they were not completely consistent, and I would like the authors to explain that better in their report of these results. For example, CarE seems to relative be less expressed using qRT-PCR, while GST increased, while comparing with their respectively results using RNA-Seq. Please explain.
Line 509: Where the host plant used the same for all the experiments? Was the age of the plant also taking into consideration as also as the plants grow their metabolites can also change. If relevant this topic is relevant, could the authors add this into the discussion?
Author Response
Dear reviewer:
We provide response point-by-point. Please see the attachment. Thank you for your comments.

Reviewer 2 Report
Comments and Suggestions for Authors
In this interesting manuscript, the authors exploited local plants regarding the habitat of the lepidopteran Aglais urticae with the aim to assess feeding preferences of larval instars. Additionally, enzymatic assays relative to different plants were also performed.Moreover, transcriptomes of A. urticae reprensenting the whole life cycle of this holometabolous insect were produced and sequenced in order to identify gene expression related to development as well as detoxification mechanisms of A. urticae given the potential toxicity of plant components to the host. The work is very well presented, methods are adequate and nicely described as well as figures composing the report. Considering the insect adaptation to feeding from four plant species, the experimental assay identified Urticae cannabina as preferred by A. urticae. Molecular results have raised expression data that (1) suggest a crosstalk between developmental and detoxifying gene expression and (2) could ideally be extended to feeding preferences of other phytophagous insects.
There is a very interesting observation made by the authors in the discussion (lines 506-508). Could you exploit with more detail the interaction between plant and insect you described?
I have only a few observations concerning grammar: In Mat. Met. (lines 198-201) and also line 502 I could not understand exactly what the authors meant. Please consider rephrasing of that sentence. In the Result titles (lines 222 and 232) please consider replacing "Analyze" for either Analysis or Analyses.
Comments on the Quality of English LanguageMinor changes are required.
Author Response
Dear reviewer:
We've answered the able comments point-to-point, so thanks again for your comments. Please see the attachment.

Reviewer 3 Report
Comments and Suggestions for Authors
This study demonstrates that U. cannabina attracts A. urticae larvae through scent, as confirmed by larval odor selection behavior bioassay. The larvae exhibit a feeding preference for U. cannabina leaves, altering the activity of detoxifying enzymes GST and CarE. Transcriptome sequencing reveals 77,624 unigenes, identifying 34 genes associated with detoxification enzymes. Notably, the study highlights genes like UDP-glucose 4-epimerase, Glutathione S-transferase, Carboxylesterase, Acetylcholinesterase, Cytochrome P450, ATP-binding cassette, and antioxidant genes. qRT-PCR validation aligns with RNA-seq results. Larvae exhibit a strong preference for U. cannabina, emphasizing its impact on A. urticae development, while detoxification gene expression peaks in larvae. Minor comments are marked on the pdf file while some major concerns are listed below.
- Major concerns and considerations: Firstly, the clear conclusion section needs to be added after discussion to conclude the results in wider and ecological context. Further, authors are requested to clarify the significance of the identified genes in the context of detoxification and A. urticae development.
- Secondly, Consider providing more context on the potential ecological implications of the observed larval preferences and enzyme activities.
- Lastly, Although the discussion on detoxification mechanisms and genetic data, while informative, lacks practical implications or applications. Authors are requested to provide blue print for future research to address these limitations by expanding the scope of plant species studied and providing practical implications for pest control strategies.

Marked on the pdf file!
Author Response
Dear reviewer:
Thank you for your comments. Below is my point-to-point response.
- Summary
Thank you very much for taking the time to review my manuscript. The three main issues we've answered that in part 2 below. Then the detailed responses, please see the pdf file (Response to reviewer 1 December), Please see the attachment. Answers to all questions are labeled in revised mode in the revised manuscript. We hope that the responses to your comments are compliant and clearly explained.
- Point-by-point response to Comments and Suggestions for Authors
Comments 1: Major concerns and considerations: Firstly, the clear conclusion section needs to be added after discussion to conclude the results in wider and ecological context. Further, authors are requested to clarify the significance of the identified genes in the context of detoxification and A. urticae development.
Response 1: We gratefully appreciate for your comment, we think that's really important. We have revised the text to increase your concerns and hope that it is now clearer. Added in the revised manuscript (lines 590-594, lines 630-639 and lines 668-673; pages 17-18). Thank you.
Comments 2: Secondly, consider providing more context on the potential ecological implications of the observed larval preferences and enzyme activities.
Response 2: The potential ecological implications, we also add new information to the revised manuscripts. Please see lines595-600, page 17. Thanks again for your advice.
Comments 3: Lastly, Although the discussion on detoxification mechanisms and genetic data, while informative, lacks practical implications or applications. Authors are requested to provide blue print for future research to address these limitations by expanding the scope of plant species studied and providing practical implications for pest control strategies.
Response 3: Thank you very much for your suggestion, we found that the article for the next step and the practical value of the application was not written in the text, so we made the addition. Please see lines 682-695, page 19.
- Response to Comments on the Quality of English Language
With respects to the English writing issue, we have already refined and revised the language in the manuscript. We hope that it will fit the requirements. Please see the revised manuscript.
We tried our best to improve the manuscript and made some changes in the manuscript. These changes will not influence the content and framework of the paper. And here we did not list the changes but changes in revised manuscripts with revision mode. Special thanks to you for your good comments and suggestions.

Round 2
Reviewer 3 Report
Comments and Suggestions for Authors
Manuscript largely improved over revision. But there persist some language and grammar issues which should be taken care off. I have marked some on the PDF file attached but feel like this should be carefully edited for it.

The manuscript requires some serious editing for the language and grammar. I have marked some of the pdf file attached to authors.
Author Response
Dear reviewer:
We have made the modifications you suggested. Please see the attachment. Thank you.
Kind regards,
Hongying Hu, Ouyan Xi, Wentao Gu
